# Compassionate care intervention for hospital nursing teams caring for older people: a pilot cluster randomised controlled trial

Lisa Jane Gould,[1] Peter Griffiths,[1,2] Hannah Ruth Barker,[1] Paula Libberton,[1] Ines Mesa-Eguiagaray,[1] Ruth M Pickering,[1] Lisa Jane Shipway,[1] Jackie Bridges[1,2]

[1]Faculty of Health Sciences, University of Southampton, Southampton, UK
[2]National Institute for Health Research Collaboration for Leadership in Applied Health Research and Care (NIHR CLAHRC) Wessex, Southampton, UK

**Correspondence to**
Dr Lisa Jane Gould;
l.j.gould@soton.ac.uk

## ABSTRACT

**Objective** Compassionate care continues to be a focus for national and international attention, but the existing evidence base lacks the experimental methodology necessary to guide the selection of effective interventions for practice. This study aimed to evaluate the Creating Learning Environments for Compassionate Care (CLECC) intervention in improving compassionate care.

**Setting** Ward nursing teams (clusters) in two English National Health Service hospitals randomised to intervention (n=4) or control (n=2). Intervention wards comprised two medicines for older people (MOPs) wards and two medical/surgical wards. Control wards were both MOPs.

**Participants** Data collected from 627 patients and 178 staff. Exclusion criteria: reverse barrier nursed, critically ill, palliative or non-English speaking. All other patients and all nursing staff and Health Care Assistant HCAs were invited to participant, agency and bank staff were excluded.

**Intervention** CLECC, a workplace intervention focused on developing sustainable leadership and work-team practices to support the delivery of compassionate care. Control: No educational activity.

**Primary and secondary outcome measures** Primary—Quality of Interaction Schedule (QuIS) for observed staff–patient interactions. Secondary—patient-reported evaluations of emotional care in hospital (PEECH); nurse-reported empathy (Jefferson Scale of Empathy).

**Results** Trial proceeded as per protocol, randomisation was acceptable. Some but not all blinding strategies were successful. QuIS observations achieved 93% recruitment rate with 25% of patient sample cognitively impaired. At follow-up there were more total positive (78% vs 74%) and less total negative (8% vs 11%) QuIS ratings for intervention wards versus control wards. Sixty-three per cent of intervention ward patients scored lowest (ie, more negative) scores on PEECH connection subscale, versus 79% of control. This was not a statistically significant difference. No statistically significant differences in nursing empathy were observed.

**Conclusions** Use of experimental methods is feasible. The use of structured observation of staff–patient interaction quality is a promising outcome measure inclusive of hard to reach groups.

**Trial registration number** ISRCTN16789770.

### Strengths and limitations of this study

► Findings from this pilot trial make an important contribution to the evidence base on the evaluation of compassionate care interventions, particularly the measurement of patient-based outcomes with older patient groups.

► This study demonstrates that use of experimental method in this field is feasible.

► The study demonstrates where blinding was effective, and where it was more difficult in a pragmatic hospital-based study.

► Only six wards were included in this study, meaning the results are not generalisable.

► The study is of insufficient scale to draw meaningful conclusions about Creating Learning Environments for Compassionate Care's effectiveness. The findings indicate, however, that more definitive evaluation is merited.

## INTRODUCTION

Healthcare systems internationally are challenged by the provision of optimal care to an ageing population.[1] Research into outcomes for older people admitted to hospital is far from encouraging with hospitalised older people at significant risk of functional decline[2] and older patients with frailty at increased risk of mortality and readmission.[3] A recent systematic review on outcomes for older people in acute care suggests there is an 'urgent need for the development and evaluation of effective interventions… that optimise the care outcomes of older patients'.[4] This review found personalised treatment plans and clear communication strategies can reduce readmission and mortality.[4] This study aims to pilot an intervention aimed at improving compassionate hospital care for older people.

Research indicates that the quality of relationships with staff is key to shaping older people's hospital experiences, with

older people valuing being seen as people, listened to and involved in treatment.[5] However, evidence from English National Health Service (NHS) and international reports[1 6–8] indicates that older people frequently fail to experience positive and caring staff attitudes and behaviours, resulting in a perceived lack of compassion. Expressed simply, compassion is 'a deep awareness of the suffering of another coupled with the wish to relieve it'.[9] There are four key components to the narrative of nursing compassion.[10] The first focuses on ideas about the moral attributes of a 'compassionate' nurse, including wisdom, humanity, love and empathy. These moral attributes are expressed through a kind of situational awareness in which vulnerability and suffering are perceived and acknowledged. These perceptions underpin participation of the nurse in responsive action that is aimed at relieving suffering and ensuring dignity, and which involves the nurse in a participatory relationship in which the nurse exercises relational capacity through which empathy is experienced and a caring pastoral relationship is constructed.[10 11]

The apparent need to improve compassionate hospital care for older people has led to the development of a number of interventions, but there is a lack of evidence for their efficacy, with utility limited by a seeming reluctance to use rigorous experimental methods for evaluation. A recent systematic review of evidence for compassionate nursing care interventions found that most of the 24 studies identified used uncontrolled before and after designs, with just four using randomised controlled designs.[10] Studies tended to be single site and small scale. A wide range of outcome measures were deployed between the studies including staff-based outcomes (eg, empathy), patient-based outcomes (eg, mood) and care outcomes (eg, patient-centredness), indicating a lack of consensus in the field as to appropriate compassionate care outcomes and how to measure them. While most studies (79%) reported a positive effect in relation to one or more outcomes, higher quality studies were less likely to report positive effects and no interventions were evaluated more than once. Thus the quality of the evidence for effectiveness in this field is predominantly low, hampered by a lack of experimental research of sufficient scale.

Responding to an absence of high-quality evidence for the effectiveness of compassionate care interventions for older patients, the study reported here aimed to pilot the use experimental methodology to evaluate a compassionate care intervention targeted at work teams in acute care settings. We aimed to provide an evidence base to guide future trial design and implementation, including feasibility of ward-level randomisation, selection of outcome measures including success in blinding, sample size calculation, minimising contamination between experimental and control clusters and maximising participation of older patients.

## METHODS

As part of a wider feasibility study, a multisite pilot cluster randomised controlled trial was undertaken with randomisation of staff and patients at ward nursing team level.[12] Medical and surgical wards with high proportion of older patients were eligible. Six wards in two NHS hospital Trusts in England were enrolled and allocated to intervention (n=4) or control (n=2). The number of clusters was determined by funding availability and the plan to run the study in at least two hospital organisations, and at least two ward specialties. Randomisation of clusters was undertaken using the ralloc command in Stata (Release 12, StataCorp) by the team statistician (IM-E) blinded to hospital and ward information other than ward specialty. Randomisation was stratified by hospital and by ward type: medicine for older people (MOP) or not MOP. The allocation was communicated to the chief investigator (JB) who oversaw its implementation in practice.

The Creating Learning Environments for Compassionate Care (CLECC) intervention is based on workplace learning theory with the ward conceptualised as a learning environment and ward team as a community of practice.[13] It is an educational programme focused on developing manager and team practices at a group level that create an expansive learning environment, theorised to enhance team capacity to provide compassionate care.[14] Expansive (rather than restrictive) environments foster workplace learning and the integration of personal and organisational development.[15–17] The intervention aims to embed ward-based manager and team practices including dialogue, reflective learning and mutual support. Research suggests that embedding such practices leads to a longer-term period of service improvement and sustainable improvements in practice.[18] CLECC training consisted of key activities, such as: monthly ward leader action learning sets; team learning activities, including local team climate analysis and values clarification; peer observations of practice and feedback to team by volunteer team members; team study days focused on team building and understanding patient experiences; mid-shift 5 min team cluster discussions; and two times weekly team reflective discussions. A Practice Educator led these activities through a 4-month implementation period, aiming to develop a team-learning plan that included measures for continuing to support leader and team practices that underpin the delivery of compassionate care beyond the initial programmed activities. Usual practice continued on control wards. Further detail on the theory and development of the CLECC intervention can be found in Bridges and Fuller.[14]

Outcome measures were assessed at baseline (2 months before intervention and prior to randomisation to groups) and follow-up (4 months after completion of CLECC implementation period). Given anticipated patient and staff turnover between assessment periods, follow-up was at cluster level rather than individual participant level, and so recruitment for baseline and follow-up assessment periods was independent. There is no single validated

measure for compassionate care, the systematic review cited above identifying 18 different types of outcome measure (a total of 67 individual outcome measures) for compassionate nursing care.[10] The most commonly used nurse-based measure identified in the review was empathy, with other measures including compassion, caring and well-being, including burnout and stress. Patient-based measures focused on overall satisfaction, quality of life, mood, agitation and well-being. Of measures that focused more on care quality, most studies used measures of the quality of interaction between nurses and patients. We chose to assess the performance of three complementary core outcomes: researcher-rated observations of the quality of staff–patient interactions, patient-reported evaluations of emotional care and nurse-reported measures of empathy. Baseline and follow-up data were also gathered on individual and ward team characteristics including patient age, cognitive impairment, ward leadership and staff turnover. We aimed to maximise the participation of older people with cognitive impairment and communication difficulties through recruitment procedures that optimised capacity to make decisions about taking part in the study.[12] Because there is insufficient literature to guide the recruitment of these groups, it was not possible at the outset to predict sample size. Instead, more flexible target recruitment rates were used.

The quality of staff–patient interactions was assessed using the Quality of Interactions Schedule (QuIS), a time sampling tool that measures the volume and quality of interactions through observation.[19] Staff–patient interactions are rated as positive social, positive care, neutral, negative protective or negative restrictive. Earlier piloting work has established its validity and reliability in acute settings.[20]

All adult patients on participating wards were assessed for eligibility to be included in observations. Patients were excluded if they were unable to communicate their choices about taking part in the research and a consultee could not be contacted. We also excluded patients who were unconscious or where there were clinical concerns (critically ill, in receipt of palliative care, high infection risk). The patient sample for observations was determined by randomisation of eligible patients, whereby a random number generator indicated the index patient for approach. Index patients were informed about the planned observations and if they agreed the observation could proceed, other eligible patients in the researcher's field of view were approached for inclusion. If the index patient declined to take part, another index patient was randomly selected, and approached as before. Study records were audited to ensure that allocation determined by randomisation was implemented in practice. Staff were informed about observations with the option to withdraw if preferred. All interactions between patients and staff were directly observed by a single researcher for 2 hours and coded (there were 10×2 hours observation sessions per ward per 3-week assessment period). Observation sessions were randomly sampled over 3 weeks from

Monday to Friday, 8.00 a.m.–10.00 p.m., and balanced between wards and time of day. Twelve researchers were trained (4 hours classroom and 6 hours field) to undertake observations.

Patient-reported evaluations of emotional care were measured using the Patient reported Evaluation of Emotional Care in Hospital (PEECH) survey tool which is validated for use in English hospital settings.[21] Designed to measure patient views on the nature of interpersonal interactions with hospital staff and patient-reported assessment of the extent to which therapeutic emotional care has occurred, the subscales are security, knowledge, personal value and connection. PEECH is sensitive to changes in service quality and in ward environment.[22] All eligible patients on the ward were invited to complete a questionnaire. Patients were excluded if there were clinical concerns or if they lacked capacity to consent. If recruited, patients were offered help by the researcher in completing the questionnaire.

Nurses' self-reported empathy was measured using the Jefferson Scale of Empathy (JSE) (Physician/HP version), a 20-item inventory in a seven-point Likert-type format ranging from Strongly Disagree to Strongly Agree with higher scores reflecting a more empathic orientation.[23] The JSE was developed and validated for use by healthcare workers, the scale is sensitive to changes in individual empathy over time and context.[24 25] All nursing staff (registered nurses and healthcare assistants) were invited to complete a questionnaire, based on a staff list supplied by the ward manager. Questionnaires in individually named envelopes were distributed by ward managers and returned via an on-ward postbox.

A number of measures were employed to enable allocation concealment and blinding. Clusters were randomly allocated to group following baseline data collection. At follow-up, researchers conducting observations were blinded to allocation, but researchers gathering questionnaire data were aware of ward allocation. It was not possible to conceal allocation from ward team nursing staff. Patients were not informed of allocation.

All analyses were carried out on an intention to treat basis. Descriptive statistics were used to show the proportion of participants that consented to participate in study. The proportion of QuIS interactions rated for each of the five categories was analysed and the frequencies of patients with the lowest (most negative) scores for each subscale were calculated. The differences between groups were tested using $x^2$ test. A three-level mixed-effects logistic regression model was fitted to investigate the effect of the CLECC intervention on the likelihood of a negative interaction. Predictive factors were included as fixed effects and presented as ORs with 95% CI, after adjustment for baseline and ward consecutively. Mean PEECH and JSE scores were calculated by subscale and in total, and differences between groups at follow-up were tested using Mann-Whitney U test. In order to determine the appropriate approach for analysis and the design effect when calculating the required sample in a

definitive trial, estimates of intracluster correlation (ICC) were generated for each outcome measure.

A small patient and public involvement (PPI) group and PPI representatives on the Steering Group oversaw and advised on intervention development, study design, selection of outcome measures and research team training.

## RESULTS

Six out of seven nursing ward managers invited to take part agreed to randomisation to either intervention or control. Three wards were recruited in each Trust, and all wards remained in the study until it closed. The wards had between 28 and 32 beds and mean patients stays ranged from 6 to 19 days. Data were collected between March 2015 and March 2016. Procedures for allocation concealment and blinding proceeded as planned, with the exception of two researcher observers at follow-up

reporting that they learnt of ward allocation from ward staff. No staff audited following observations reported that their behaviour had changed because they were being observed. Researcher field notes reflect reports from hospital managers that discussions about CLECC between staff on intervention and control wards had the potential to influence practice on the control wards, but we did not detect evidence of contamination.

### Participant flow

Figure 1 shows the flow of clusters and participants through the pilot trial. Randomisation took place after baseline data collection, but results are presented by allocation for baseline and follow-up data to enable comparisons between groups.

For staff–patient observations, figure 1 illustrates the number of approaches rather than individual patients, as some patients were invited more than once to be

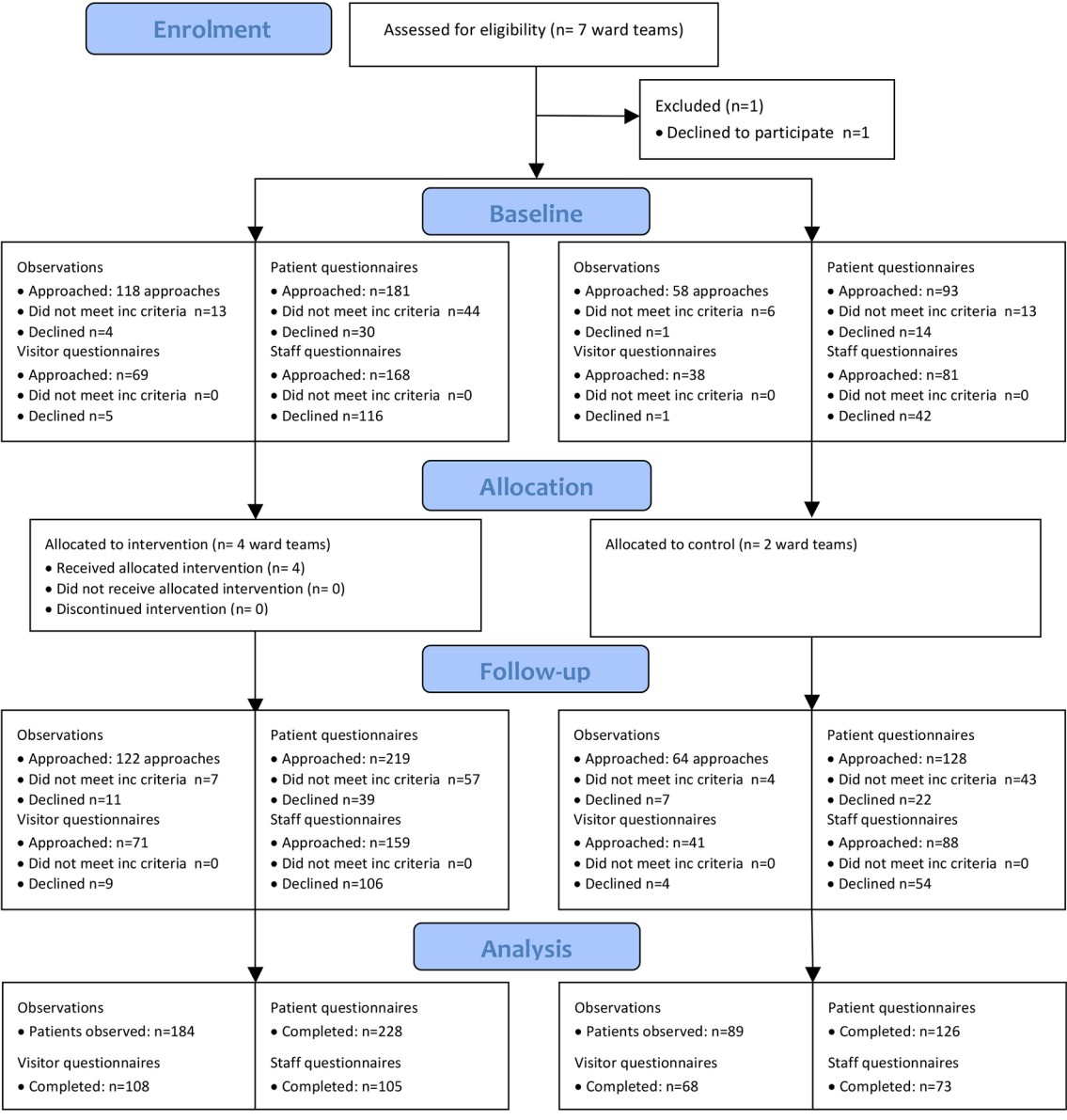

**Figure 1** Consolidated Standards of Reporting Trials flow diagram. inc, inclusion.

**Table 1** Patient characteristics

| Variable | |
|---|---|
| **Observations (n=273), missing data=0** | |
| Age | |
| 18–30 years | 1 (0%) |
| 31–40 years | 2 (1%) |
| 41–50 years | 7 (3%) |
| 51–60 years | 14 (5%) |
| 61–70 years | 14 (5%) |
| More than 70 years | 235 (86%) |
| Gender | |
| Male | 63 (23%) |
| Female | 210 (77%) |
| Cognitive impairment | |
| Yes | 68 (25%) |
| No | 205 (75%) |
| **Questionnaires (n=321), missing data=33** | |
| Age | |
| 18–30 years | 4 (1%) |
| 31–40 years | 3 (1%) |
| 41–50 years | 9 (3%) |
| 51–60 years | 15 (5%) |
| 61–70 years | 24 (7%) |
| More than 70 years | 266 (83%) |
| Gender | |
| Male | 95 (30%) |
| Female | 226 (70%) |
| Cognitive impairment | |
| Yes (n=43) 12% | |
| No (n=315) 88% | |

involved. Recruitment rate for observations at baseline was 97% (152 out of 157 approaches to eligible patients), and at follow-up was 90% (157 out of 175). Recruitment rates were similar between intervention and control wards (96% vs 98% at baseline, 90% vs 88% at follow-up). Twenty-three participants declined to participate for reasons including 'not feeling up to it' (17%), or 'too unwell' (4%). No specific reason was recorded for 70%. In 17% (63 out of 362 approaches) patients were assessed as not having capacity to make the decision to take part. In 67% (42 out of 63) of these occasions, researchers were able to contact a consultee for advice and in 100% of these cases the consultee advised that the patient should participate. A final 273 patients were observed (133 at baseline and 140 at follow-up). The mean age of patients observed was 82 years (84 years in intervention group and 77 in control) (table 1). Most patients were female (77%) and 25% had evidence of cognitive impairment, with no significant differences by experimental group. All observation data gathered were included in analysis.

Across both assessment periods, 77% (359 out of 464) of eligible patients agreed to take part in the questionnaire survey. Overall recruitment rates were similar between intervention and control wards (77% vs 78%). Most frequent reasons recorded for patients declining participation in the questionnaire survey were 'tired' (40%, n=12) and 'questionnaire too difficult' (10%, n=3). The most frequent reasons recorded for excluding patients were 'not having capacity' (43%, n=48) and 'very cognitively impaired' (29%, n=32). Ninety-nine per cent (354 of 359) of patients who consented returned a completed questionnaire, with researchers helping with completion in 68% of cases. Most patients were female (70%), and aged over 70 years (83%). Twelve per cent of patient questionnaires were completed by patients with cognitive impairment. Intervention group patients completing questionnaires at baseline included a higher proportion of younger patients (22% aged ≤60 years vs 0%) and of males (43% vs 25%). There were no other notable differences by experimental group (table 1).

Of 496 questionnaires distributed to nursing staff, 36% (n=178) were completed and returned (37% at baseline, 35% at follow-up). Baseline return rates were lower on intervention wards (31% vs 48%), but at follow-up were more similar between experimental groups (33% vs 39%). Most staff who returned a completed questionnaire were female (87%) and median age group was 26–35 years. Questionnaires were returned by 74 healthcare assistants (42%), 74 staff nurses (42%), and 18 sisters/charge nurses (10%), (missing data=6%). There were no notable differences in job role by experimental group. All returned questionnaires (91 at baseline and 87 at follow-up) were included in analyses.

### Baseline and outcome measures
As planned, 120 hours of observations took place in each assessment period, resulting in data collected on 3109 interactions between staff and patients over 240 hours. On average, each patient had six interactions with hospital staff per hour. Most interactions were rated as positive care (59%) and least interactions as negative protective (4%) for each experimental group at both assessment periods (table 2).

At follow-up, there were more total positive (positive social and positive care) and less total negative (negative protective and negative restrictive) scores for intervention wards than control (78% vs 74%, 8% vs 12%). $x^2$ testing suggested these differences were significant (P=0.017). However, multilevel logistic regression results indicate that once other variables are taken into account, the odds of a negative interaction are not significantly reduced because of the effect of the CLECC intervention (table 3). Results are in the direction of an effect favourable to CLECC, that is, there were less negative interactions on intervention wards, but this is not a statistically significant difference (adjusted OR 0.30 (95% CI 0.07 to 1.32)).

**Table 2** Quality of staff–patient interaction QuIS by experimental group (baseline and follow-up)

| QuIS rating | Baseline (n=1554) | | Follow-up (n=1555) | |
|---|---|---|---|---|
| | CLECC (n=1143) | Control (n=411) | CLECC (n=1119) | Control (n=436) |
| Positive social | 167 (15%) | 37 (9%) | 243 (22%) | 64 (14%) |
| Positive care | 672 (59%) | 255 (62%) | 632 (57%) | 260 (60%) |
| Neutral | 190 (17%) | 77 (19%) | 151 (14%) | 62 (14%) |
| Negative protective | 42 (4%) | 17 (4%) | 36 (3%) | 21 (5%) |
| Negative restrictive | 72 (6%) | 25 (6%) | 57 (5%) | 29 (7%) |

CLECC, Creating Learning Environments for Compassionate Care; QuIS, Quality of Interaction Schedule.

Table 4 shows the mean patient evaluations of emotional care (PEECH) values by experimental group. Higher scores indicate better patient-reported experiences. Connection subscale scores were consistently lower than on other subscales. Differences between groups at follow-up favour CLECC in total score and three of the four subscales, but these differences were not significant.

Levels of staff self-reported empathy using JSE varied across individual wards at baseline and at follow-up. There was no significant difference between groups (P=0.800).

At ward level, ICCs for QuIS, PEECH and JSE were low (<0.027). The ICC for QuIS at ward level was higher, although still small (0.071), but high at observation session level (0.411).

## DISCUSSION

This study aimed to deliver a compassionate care intervention in acute care settings, pilot the use of experimental methodology and assess the performance of selected outcome measures. We aimed to provide an evidence base to guide future trial design and implementation, including acceptability of ward-level randomisation, the feasibility of assessing outcome measures and other measures of trial implementation such as recruitment and inclusivity, sample size calculation and clustering for future trial, blinding and contamination. The high recruitment rate of ward managers on behalf of their teams and subsequent lack of attrition of any of the ward teams recruited indicate that trial randomisation and the CLECC intervention are acceptable to medical and surgical nursing teams in acute care hospitals. Recruitment processes and methods appeared to be inclusive of all nursing staff levels and of older patients. Observations, in particular, were highly acceptable to patients with an overall recruitment rates of 93%. Questionnaire response rates varied, as discussed below. Our findings suggest that the CLECC intervention may have a favourable effect in reducing negative interactions between staff and patients, and in reducing patients' experiences of lack of emotional connection with staff. However as expected, because of the scale of this pilot, there is no certainty that any apparent positive effects are not produced by chance alone, rather than the impact of the CLECC intervention.

Hospitalised older patients with cognitive impairment are a traditionally hard-to-reach group and even though they appear more prone to negative experiences of hospital care,[26] they are often excluded from research.[5 27 28] It is estimated that up to 25% of beds in acute hospitals are occupied by people with dementia, with the figure likely to be higher on specialist older people's wards.[29 30]

**Table 3** QuIS multilevel logistic regression results: ORs of a negative interaction

| Variables | Model 1 unadjusted OR (95% CI) (n=3111) | Model 2 adjusted OR (95% CI) (n=3111) | Model 3 adjusted OR (95% CI) (n=3111) |
|---|---|---|---|
| CLECC effect | 0.72 (0.35 to 1.51) | 0.47 (0.17 to 1.29) | 0.30 (0.07 to 1.32) |
| Time period (baseline vs follow-up) | | 0.56 (0.22 to 1.43) | 0.38 (0.11 to 1.32) |
| Ward | | | |
| A | | | 1.00 |
| B | | | 0.60 (0.20 to 1.83) |
| C | | | 0.80 (0.21 to 3.05) |
| D | | | 0.75 (0.24 to 2.35) |
| E | | | 0.61 (0.19 to 1.90) |
| F | | | 0.23 (0.05 to 1.02) |
| Variance component estimates (95% CI) | | | |
| Observation session level (n=120) | 2.13 (1.25 to 3.62) | 2.09 (1.23 to 3.55) | 1.96 (1.14 to 3.37) |
| Patient level (n=273) | 0.51 (0.23 to 1.13) | 0.51 (0.23 to 1.13) | 0.51 0.23 to 1.13) |

CLECC, Creating Learning Environments for Compassionate Care; QuIS, Quality of Interaction Schedule.

**Table 4** PEECH mean (SD) scores by experimental group (baseline and follow-up)

| PEECH mean (SD) | Baseline (n=168) | | Follow-up (n=186) | | P value |
|---|---|---|---|---|---|
| | CLECC (n=105) | Control (n=63) | CLECC (n=123) | Control (n=63) | |
| Security (0–3) | 2.48 (0.55) | 2.36 (0.51) | 2.48 (0.50) | 2.46 (0.48) | 0.653 |
| Knowing (0–3) | 2.18 (0.82) | 2.30 (0.72) | 2.19 (0.88) | 2.26 (0.66) | 0.800 |
| Personal value (0–3) | 2.34 (0.57) | 2.35 (0.58) | 2.43 (0.57) | 2.31 (0.57) | 0.071 |
| Connection (0–3) | 1.68 (0.74) | 1.61 (0.84) | 1.81 (0.82) | 1.71 (0.63) | 0.350 |
| Total PEECH score (0–66) | 49.2 (11.5) | 48.4 (12) | 50.6 (11.3) | 48.5 (9.8) | 0.116 |

CLECC, Creating Learning Environments for Compassionate Care; PEECH, Patient reported Evaulation of Emotional Care in Hospitals.

While cognitive deficits may limit some people's ability to share their experiences, our study has been successful in devising recruitment and data collection methods that maximise their inclusion. Overall 25% of patients observed in this study had evidence of cognitive impairment, suggesting a sample representative of the wider hospital population. Twelve per cent of patient questionnaires returned were completed by patients with cognitive impairment, indicating the questionnaire method was less inclusive than observation methods. Participating in an observation does not require any particular state of health, abilities or performance form the patient in question, whereas participating in a questionnaire about one's care experiences requires a minimum orientation to place, language skills and attention.[28] In addition, using questionnaire methods may be psychologically threatening to patients still in receipt of care, regardless of cognitive status.[31]

The validity of observer ratings as accurate representation of patient experiences merits attention. Because main study observation and questionnaire data were gathered from different patient groups, it was not possible to test the validity of observer ratings against patient-reported experience. However, in earlier piloting work we found 79% agreement (weighted kappa 0.40: P<0.001; indicating fair agreement) between patients' and observers' ratings of interaction quality.[20] Our earlier work did not include people with a cognitive impairment and validation of QuIS ratings with this patient group may be a necessary next step in the tool's development. In addition, if the proportion of negative interactions is the primary outcome measure in a future study, understanding which interactions are rated by observers (and, where possible, patients) as negative, and why, is an important next step, as is working with patient representatives to establish their views on the size of a meaningful reduction in negative interactions. Further study can also be used to develop more effective procedures to blind observers from experimental allocation in advance of an experimental study. In addition, the high ICC we found at an observation session level merits the exploration of the cause of this variance and the feasibility of different approaches to data collection that reduce its impact, for instance, shorter observation sessions. Our findings echo those of Goldberg and Harwood[27] that structured non-participant observation appears to be the most promising method to describe the experiences of older people with cognitive impairment in the general hospital setting, and so further evaluation and testing of QuIS across these parameters would be a valuable foundation to its further use as an outcome measure in acute settings.[27]

While the response rate to patient questionnaires was good (77%), of all the patient questionnaires returned, just 12% were completed by patients with cognitive impairment. While questionnaires provide an opportunity for patient to directly rate their care, less successful recruitment of a group known to be vulnerable to more negative experiences in hospital, means that any results may not be a valid representation of this group's experiences. The response rate to nursing questionnaires was low (36%), with some larger scale studies showing response rates of European nurses to be 62%, and US nurses to be around 39%.[32] Improving staff survey response rates through further feasibility work would improve confidence that conclusions in empathy levels across staff groups can be drawn with more confidence.

This study was piloted on a small number of wards in two hospitals so the findings are not generalisable. In addition, being observed could, in itself, change staff behaviours, and a common limitation of trials of this kind when it is not possible to conceal allocation from staff, is that bias may influence staff responses to observations and questionnaires. Additionally the finding of possible contamination between wards means that intervention and control conditions should not run in the same organisation over the same time period.

Findings from our wider study, reported elsewhere, that implementation of the CLECC intervention was uneven between wards, difficult to sustain and dependent on organisational support,[33] indicate that, while experimental research in this field is necessary, it will not provide sufficient explanation of results if conducted in isolation. However, the findings reported here represent valuable groundwork to the further development of sound experimental design in a field in which good design and implementation are very much needed.

**Acknowledgements**  The authors thank all patients, research nurses and staff in the participating hospitals. Without their support, the study would not have been possible. With thanks to our fellow investigators not listed as authors: Rosemary Chable, Emma Munro, Alison Fuller, James Raftery, Avan Aihie Sayer, Greta Westwood, Wendy Wigley and Guiqing Yao.

**Contributors**  LJG, HRB, LJS and PL: implementation, data collection, analysis, interpretation of results and write up of paper. JB: design, implementation, data collection, analysis, interpretation of results and write up of paper. PG, RMP: design, analysis, interpretation of results and write up of paper. IM-E: analysis and interpretation of results and write up of paper.

**Funding**  This project was funded by the National Institute for Health Research (Health Services and Delivery Research programme) (project number 13/07/48).

**Disclaimer**  The views and opinions expressed therein are those of the authors and do not necessarily reflect those of the Health Services and Delivery Research programme, NIHR, NHS or the Department of Health.

**Competing interests**  None declared.

**Patient consent**  Obtained.

**Ethics approval**  Ethical approval for the study was granted by the National Social Care Research Ethics Committee 14/IEC08/1018.

**Provenance and peer review**  Not commissioned; externally peer reviewed.

**Data sharing statement**  Data can be requested from the corresponding author.

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
