## [Reviewer comments · BMJ Open]

ARTICLE DETAILS

TITLE (PROVISIONAL)	Compassionate care intervention for hospital nursing teams caring for older people: a pilot cluster randomised controlled trial
AUTHORS	Gould, Lisa; Griffiths, Peter; Barker, Hannah; Libberton, Paula; Mesa-Eguiagaray, Ines; Pickering, Ruth; Shipway, Lisa; Bridges, Jackie

VERSION 1 – REVIEW

REVIEWER	Stephanie Tierney Warwick Medical School, UK
REVIEW RETURNED	20-Jul-2017

GENERAL COMMENTS	This paper reported on a study in an area where experimental approaches are rare. I think it raises some interesting points for reflection. However, there are aspects that I feel need revising in the current manuscript. Title • Should the authors include 'older people' in the title, given this is the population of interest? Introduction • The authors claim that the intervention is to address compassionate care. However, there is a lack of definition about what they mean by this concept and, for me, it is more about addressing relational aspects of care. Definitions of compassionate care relate to noticing, feeling and acting to alleviate suffering and distress, which is broader than relational aspects of care addressed and assessed in the current paper. Methods • I did wonder how the elements of the intervention were decided upon, and whether consideration had been given to past work in this area, such as the research by Dewar and colleagues, when designing the intervention?• Is 4 months long enough to expect a change/improvement in a ward culture?• The authors mention trying to maximise participation of patients often excluded from research – what steps did they take to ensure this happened?• Measuring compassionate care through observation is problematic, given that whether an action is experienced as compassionate or not is subjective and relates to an individual's perspective.
--

	I appreciate that measuring compassionate care is difficult (see paper by Tierney et al in Nursing Management) – but what is measured in this study seems to go back to this narrow perceptive of compassionate care, which seems limited to patient-professional interaction.  • There is the issue of asking patients to complete a questionnaire about care received when they are still in-patients when they may feel vulnerable and prompted to respond favourably. Results  • There was very low engagement from nurses in completing questionnaires – perhaps some more feasibility work is required to explore why this may be and how to increase questionnaire completion, before rolling out to a larger sample. • Figure 1 is lacking a title.
--	--

REVIEWER	Michelle Farr Dr Michelle Farr Senior Research Associate in Applied Social Science (Qualitative) Research National Institute for Health Research Collaborations for Leadership in Applied Health Research and Care West (NIHR CLAHRC West) University Hospitals Bristol NHS Foundation Trust Bristol Medical School, University of Bristol UK
REVIEW RETURNED	21-Aug-2017

GENERAL COMMENTS	The article provides an important contribution to the evaluation of compassionate care interventions, focusing on their impact on staff and patients and their interactions. The study objectives are clearly explained within the manuscript. These objectives are important to develop research and evidence in compassionate care interventions. The abstract is clear and balanced, highlighting the contribution of the research. The study design is appropriate to the objectives of the study. Clear aims that are grounded within an understanding of limits of current research. Series of clearly stated research aims. Well-designed methodology. The methods are clearly described in appropriate detail. The paper says that: ‘Baseline and follow-up data were also gathered on individual and ward team characteristics.’ It would be helpful to include the type of information gathered. Participant consent is addressed appropriately. I can’t find an ethical review statement or reference number, the box where this may be reported on p.27 lines 30-32 seems to be missing. The outcomes are clearly described in the paper. It is good that the authors have used a range of different outcome measures, as the paper highlights that there is a lack of consensus about outcomes and how to measure them accurately in this field of research. This paper provides an important contribution to this area. Statistics used are fully described. As my specialism is in qualitative research, I am less qualified to comment on the appropriateness of the statistics used. References are up to date and appropriate. The results address the objectives of the study.
---

	The data is clearly presented. The paper discusses some interesting results. It is an achievement to have involved the number of participants with cognitive impairments that they have. The discussions and conclusion are justified by the results. The paper contributes to the development of appropriate methods to research experiences of care with older people with complex needs. The study's limitations are well written but could be developed further. The low staff survey response rates made me wonder how much staff were engaged with the intervention. There is no mention of any qualitative process evaluation in the paper but looking at the protocol I see that a qualitative process and economic evaluation were also carried out as part of the study. I presume that these will be reported in other papers that the authors will be writing, however it would be useful to explain that these evaluations were carried out in the main paper, and are reported separately. This will provide the reader with a more complete understanding of how the experimental methodology described, was part of a broader study. This is important because if an objective is to 'to guide the selection of effective interventions for practice', we need more than an experimental methodology as described in the paper. At the moment this paper leaves a black box question of what's going on in the intervention, and why is it working or not? How is it being implemented? Did leaders and staff like the intervention or not? How influential were ward climates in the implementation of the intervention? The wider study described in the protocol addresses all these issues. I would recommend including a short acknowledgement of these issues in the main paper, explaining that the broader study addressed these wider questions, referencing the protocol, so that readers know where to get further information if they are interested in these broader issues as well. p.13 lines 16-18 Is there an additional limitation here? Staff may know whether they have been part of the CLECC intervention, as they may have participated in it. Could this have affected the way that intervention staff behaved in comparison with non-intervention staff when being observed? 'Allocation concealment and blinding proceeded as planned, with the exception of two (out of eight) researcher observers at follow-up reporting that they learned of ward allocation from ward staff.' Two researchers may have experienced some unconscious bias as two out of eight learned of ward allocation from staff. May be worth adding to limitations? The CONSORT checklist is complete, apart from a box in the 4th column on p.27 lines 30-32 seems to be missing, with details of ethical approval. Please add details. Standard of English is good, it is a well explained and coherent paper. However, I did find the following occasions where there may be typos, or sentences are unclear. p. 2 line 19: participate, not participant? p.5 line 49 Impact was assessed (typo) across three complementary core outcomes p.6 line 47 All interactions between patients and staff directly observed for two hours and coded Include were? p.6 line 50-51 over a three weeks over three weeks, or a three week period? p.8 line 53 declined to participation declined to participate? p.11 line 20 ICC not given in full – ICCs p.12 line 53 A limitation to this study is that has only been piloted on Word missing?
--	--

	p.13 line 47 this study allows it to addresses global concerns for address?
--	---

VERSION 1 – AUTHOR RESPONSE

R1

Title

Should the authors include 'older people' in the title, given this is the population of interest?

Response: Thank you, we will add this

R1

Introduction

The authors claim that the intervention is to address compassionate care. However, there is a lack of definition about what they mean by this concept and, for me, it is more about addressing relational aspects of care. Definitions of compassionate care relate to noticing, feeling and acting to alleviate suffering and distress, which is broader than relational aspects of care addressed and assessed in the current paper.

Response: Thank you, we have rewritten the paragraph to better clarify our use of the term "compassionate care", including a definition.

R1

I did wonder how the elements of the intervention were decided upon, and whether consideration had been given to past work in this area, such as the research by Dewar and colleagues, when designing the intervention?

Response: Limited word count has meant that we cannot share detail of how the intervention was developed, including how literature review informed its development and focus. We have however now directed readers to a published paper where they can access this information.

R1

Is 4 months long enough to expect a change/improvement in a ward culture?

Response: Thank you, this is an excellent point. There is no published evidence to guide what would be an effective time period. We have reported that the original implementation period was 4 months but that this was expected to lead to a longer term period of innovation and improvement. Hence the decision to measure follow-up at 8 months following the start of the original implementation period. We have clarified information about the implementation period in the Methods section.

R1

The authors mention trying to maximise participation of patients often excluded from research – what steps did they take to ensure this happened?

Response: We could write a whole paper on this (!) but are too limited for space in this paper to provide much detail. We have added some overview information and added a reference to the study protocol so this information can be accessed.

R1

Measuring compassionate care through observation is problematic, given that whether an action is experienced as compassionate or not is subjective and relates to an individual's perspective. I appreciate that measuring compassionate care is difficult (see paper by Tierney et al in Nursing Management) – but what is measured in this study seems to go back to this narrow perceptive of compassionate care, which seems limited to patient-professional interaction.

Response: We agree that measurement of such a complex concept is problematic, hence our approach to testing three complementary measures. The value of an observation based measure is that, as we demonstrated, it is possible to be more inclusive of people with cognitive impairment. In the Discussion, we have addressed the validity of using an observation-based measure to infer people's experiences, and cite some positive findings from earlier pilot work. We have re-ordered the relevant parts of the Discussion and added more information to address this reviewer's concern.

We note Reviewer 2's comments: "It is good that the authors have used a range of different outcome measures, as the paper highlights that there is a lack of consensus about outcomes and how to measure them accurately in this field of research. This paper provides an important contribution to this area."

R1

There is the issue of asking patients to complete a questionnaire about care received when they are still in-patients when they may feel vulnerable and prompted to respond favourably.

Response: We have included reference to this issue in the Discussion section

R1

There was very low engagement from nurses in completing questionnaires – perhaps some more feasibility work is required to explore why this may be and how to increase questionnaire completion, before rolling out to a larger sample.

Response: Thank you for the suggestion. We have included this in the Discussion.

R1

Figure 1 is lacking a title.

Response: We have added a title

R2

The article provides an important contribution to the evaluation of compassionate care interventions, focusing on their impact on staff and patients and their interactions. The study objectives are clearly explained within the manuscript. These objectives are important to develop research and evidence in compassionate care interventions.

The abstract is clear and balanced, highlighting the contribution of the research.

The study design is appropriate to the objectives of the study. Clear aims that are grounded within an understanding of limits of current research. Series of clearly stated research aims.

Well-designed methodology. The methods are clearly described in appropriate detail.

Participant consent is addressed appropriately.

The outcomes are clearly described in the paper. It is good that the authors have used a range of different outcome measures, as the paper highlights that there is a lack of consensus about outcomes and how to measure them accurately in this field of research. This paper provides an important contribution to this area. Statistics used are fully described. As my specialism is in qualitative research, I am less qualified to comment on the appropriateness of the statistics used.

References are up to date and appropriate.

The results address the objectives of the study.

The data is clearly presented. The paper discusses some interesting results. It is an achievement to have involved the number of participants with cognitive impairments that they have.

The discussions and conclusion are justified by the results. The paper contributes to the development of appropriate methods to research experiences of care with older people with complex needs.

Response: Thank you for your positive comments

R2 The paper says

that: 'Baseline and follow-up data were also gathered on individual and ward team characteristics.' It would be helpful to include the type of information gathered.

Response: Thank you – we have added some examples of the type of information.

R2 I can't find an ethical review statement or reference number, the box where this may be reported on p.27 lines 30-32 seems to be missing.

The CONSORT checklist is complete, apart from a box in the 4th column on p.27 lines 30-32 seems to be missing, with details of ethical approval. Please add details.

Response: Thank you – we have added this information.

The study's limitations are well written but could be developed further. The low staff survey response rates made me wonder how much staff were engaged with the intervention. There is no mention of any qualitative process evaluation in the paper but looking at the protocol I see that a qualitative process and economic evaluation were also carried out as part of the study. I presume that these will be reported in other papers that the authors will be writing, however it would be useful to explain that these evaluations were carried out in the main paper, and are reported separately. This will provide the reader with a more complete understanding of how the experimental methodology described, was part of a broader study. This is important because if an objective is to 'to guide the selection of effective interventions for practice', we need more than an experimental methodology as described in the paper. At the moment this paper leaves a black box question of what's going on in the intervention, and why is it working or not? How is it being implemented? Did leaders and staff like the intervention or not? How influential were ward climates in the implementation of the intervention? The wider study described in the protocol addresses all these issues. I would recommend including a short acknowledgement of these issues in the main paper, explaining that the broader study addressed these wider questions, referencing the protocol, so that readers know where to get further information if they are interested in these broader issues as well. Thank you. As suggested, we have referenced the wider study in Methods and given an overview of the findings of the process evaluation in the Discussion, together with consideration of the implications for future design.

R2

p.13 lines 16-18 Is there an additional limitation here? Staff may know whether they have been part of the CLECC intervention, as they may have participated in it. Could this have affected the way that intervention staff behaved in comparison with non-intervention staff when being observed?

Response: Thank you. We have included this as a potential limitation.

R2

'Allocation concealment and blinding proceeded as planned, with the exception of two (out of eight) researcher observers at follow-up reporting that they learned of ward allocation from ward staff.'
Two researchers may have experienced some unconscious bias as two out of eight learned of ward allocation from staff. May be worth adding to limitations?

Response: Thank you. Again, we have no evidence to suggest this, but have alluded to it as a limitation.

R2

Standard of English is good, it is a well explained and coherent paper. However, I did find the following occasions where there may be typos, or sentences are unclear.

p. 2 line 19: participate, not participant?

p.5 line 49 Impact was assessed (typo) across three complementary core outcomes

p.6 line 47 All interactions between patients and staff directly observed for two hours and coded Include were?

p.6 line 50-51 over a three weeks
over three weeks, or a three week period?

p.8 line 53 declined to participation
declined to participate?

p.11 line 20 ICC not given in full – ICCs

p.12 line 53 A limitation to this study is that has only been piloted on
Word missing?

p.13 line 47 this study allows it to addresses global concerns for address?

Response: Thank you for your thorough review. We have made corrections to all these points.

VERSION 2 – REVIEW

REVIEWER	Stephanie Tierney University of Warwick
REVIEW RETURNED	17-Oct-2017

GENERAL COMMENTS	The authors are to be congratulated on their attempt to conduct a RCT in the complex area of compassionate care. I think there needs to be a bit more in the introduction on the meaning of compassionate care; it's not just about relationships but acting to alleviate suffering. It is a word that is used without there always being a consensus around what it means. Was there any patient/public involvement in developing the study? How was randomisation performed (e.g. sealed envelopes?) Was it possible to change the allocation? This includes for the patient observations (could it be manipulated who was seen)? Were patient observations validated in anyway? Were they done by just one researcher?
--

	It was not clear how the activities described as part of CLECC were specifically addressing compassion (this goes back to the comment made above about a lack of reference in the introduction to what is meant by compassion). It was also not clear what theoretical stance was being taken in terms of addressing compassionate care – is it possible to do this at an individual level, or does it have to be done at a more institutional level? I was not sure the measures used actually measured compassion – are they more a proxy of this? The authors do allude to the difficulty of measuring compassion, but empathy is different to compassion – it could be seen as part of compassion but not the same (see Gilbert 2013) (again perhaps this needs to be alluded to in the introduction). Also the PEECH does not centre on alleviation of suffering, so therefore does it measure compassion? Was a power calculation conducted for the study, or was this part of the reason why the feasibility work was being conducted? In the discussion, the claim that CLEEC seemed acceptable to the nursing teams is not possible to state from the data provided in the paper as this is not assessed in the measures used. This also seems a questionable statement given the final paragraph of the paper. Furthermore, I don't think the authors can make a claim that CLEEC had any effect on compassionate care based on the results reported in this paper.
--	---

REVIEWER	Michelle Farr CLAHRC West, University of Bristol, UK
REVIEW RETURNED	30-Oct-2017

GENERAL COMMENTS	The authors have satisfactorily addressed all the points made by reviewers. I recommend the paper for publication. I did notice one sentence that may need clarification on p.12, lines 35-39. "Twelve percent of patient questionnaires returned were completed by patients with cognitive impairment, indicating the questionnaire method was inclusive than observation methods." Is a word missing here?
--

VERSION 2 – AUTHOR RESPONSE

R1

I think there needs to be a bit more in the introduction on the meaning of compassionate care; it's not just about relationships but acting to alleviate suffering. It is a word that is used without there always being a consensus around what it means.

Response: We have expanded our operational definition of compassion

R1

Was there any patient/public involvement in developing the study?

Response: We have added information about PPI input

R1

How was randomisation performed (e.g. sealed envelopes?) Was it possible to change the allocation? This includes for the patient observations (could it be manipulated who was seen)?

Response: We have provided further detail as requested.

R1

Were patient observations validated in anyway? Were they done by just one researcher?

Response: We have extended our discussion on the validity of QuIS observation ratings in relation to patient experience in the Discussion. We have added information to Methods that each observation was undertaken by a single researcher, and also that 12 researchers in total were involved in study observations. We have also added information about observer training.

R1

It was not clear how the activities described as part of CLECC were specifically addressing compassion (this goes back to the comment made above about a lack of reference in the introduction to what is meant by compassion). It was also not clear what theoretical stance was being taken in terms of addressing compassionate care – is it possible to do this at an individual level, or does it have to be done at a more institutional level?

Response: We have added further information and ensured that the original intervention paper is referenced which more fully describes the underlying programme theory. Space constraints mean that we are unable to provide more information here.

R1

I was not sure the measures used actually measured compassion – are they more a proxy of this? The authors do allude to the difficulty of measuring compassion, but empathy is different to compassion – it could be seen as part of compassion but not the same (see Gilbert 2013) (again perhaps this needs to be alluded to in the introduction). Also the PEECH does not centre on alleviation of suffering, so therefore does it measure compassion?

Response: Hopefully our more detailed introduction provides reassurance on the ways we have chosen to measure compassion. As noted by the reviewer, we acknowledge on p.4 “There is no single validated measure for compassionate care so its impact was assessed across three complementary core outcomes: researcher-rated observations of the quality of staff-patient interactions, patient-reported evaluations of emotional care and nurse-reported measures of empathy”. We have extended our discussion of the complexity of measuring this concept, drawing on findings from a recently published systematic review. We agree it is a problematic field hence our attempts to progress the science of measurement.

To respond to the reviewer’s further points:

On p.13-14 in the Discussion we discuss the merit of using QuIS observations as a proxy for patient experience. We have added to this discussion, including identifying some areas for future development of the tool, including validating ratings with patients with cognitive impairment (see earlier)

Measuring empathy is indeed relevant to compassion because compassion is not just about alleviating suffering, but also about a “deep awareness” of the suffering i.e. empathy with the patient. In our extended definition now included in the introduction, we identify moral attributes of a ‘compassionate’ nurse, including wisdom, humanity, love, and empathy.

In addition, these moral attributes are expressed through a kind of situational awareness in which vulnerability and suffering are perceived and acknowledged. In the review now cited on pp.4-5, we identify that empathy has been used in other studies as a compassionate care measure and is the most frequently used outcome used for this purpose in relation to nursing care.

With regard to choice of PEECH, it is a measure overall of patient views on the nature of interpersonal interactions with hospital staff and patient-reported assessment of the extent to which therapeutic emotional care has occurred, and we identified this as the most suitable measure for the study, particularly given its validation in hospital environments. The PEECH focus relates to the final parts of our definition of compassion elucidated now in the introduction: “participation of the nurse in responsive action that is aimed at relieving suffering and ensuring dignity, and which involves the nurse in a participatory relationship in which the nurse exercises relational capacity through which empathy is experienced and a caring pastoral relationship is constructed.” We have added more information about PEECH focus in the paper.

In the Discussion section we have also added some discussion about the performance of PEECH and JSE. Although it focuses more on response rates, our addition does indirectly address validity as well. Because both instruments selected are already well validated, we did not undertake any validity testing in this study.

R1

Was a power calculation conducted for the study, or was this part of the reason why the feasibility work was being conducted?

Response: We did not perform power calculation for the study, explaining on p. 3 “We aimed to provide an evidence base to guide future trial design and implementation, including feasibility of ward level randomisation, selection of outcome measures including success in blinding, sample size calculation, minimising contamination between experimental and control clusters, and maximising participation of older patients.”, and on p. 4 “Because there is insufficient literature to guide the recruitment of these groups, it was not possible at the outset to predict sample size. Instead, more flexible target recruitment rates were used.”

We have added a sentence on p.3 to explain how we decided on the number of clusters.

R1

In the discussion, the claim that CLEEC seemed acceptable to the nursing teams is not possible to state from the data provided in the paper as this is not assessed in the measures used. This also seems a questionable statement given the final paragraph of the paper. Furthermore, I don't think the authors can make a claim that CLEEC had any effect on compassionate care based on the results reported in this paper.

Response: We have added more information to clarify the basis for the claim of acceptability of the CLEEC intervention and trial randomisation. This does not therefore contradict the conclusions in the final paragraph. We agree that we cannot claim an effect and believe that we have made this already very clear:

Examples

p. 12 “However as expected, because of the scale of this pilot, there were no significant differences once other variables were accounted for” [we have reworded this for added clarity]

p.14 “This study was piloted on a small number of wards in two hospitals so the findings may not be generalizable”. [we have changed this final claim to “are not generalisable”]

There was also a sentence referring to the clear design effect of QuIS at an observation session level – this referred to the ICC findings, and the effect of the observation session on clustering – so we have reworded our passage about ICC findings in case that is what the reviewer is referring to.

R2

I did notice one sentence that may need clarification on p.12, lines 35-39. "Twelve percent of patient questionnaires returned were completed by patients with cognitive impairment, indicating the questionnaire method was inclusive than observation methods." Is a word missing here?

Response: Thank you – other changes mean that we have re-written this sentence and so the omission no longer needs correcting.

VERSION 3 – REVIEW

REVIEWER	Stephanie Tierney Warwick Medical School
REVIEW RETURNED	10-Nov-2017
GENERAL COMMENTS	I am happy with the revisions made by the authors.

VERSION 3 – AUTHOR RESPONSE

Reviewer: 1

Reviewer Name: Stephanie Tierney

Institution and Country: Warwick Medical School

Please state any competing interests or state 'None declared': None declared

Please leave your comments for the authors below

Comment: I am happy with the revisions made by the authors.

Our response: thank you very much for your careful review